# Sensitivity of Safe Trajectory in a Game Environment on Inaccuracy of Radar Data in Autonomous Navigation

**DOI:** 10.3390/s19081816

**Published:** 2019-04-16

**Authors:** Józef Lisowski

**Affiliations:** Faculty of Marine Electrical Engineering, Gdynia Maritime University, 81-225 Gdynia, Poland; j.lisowski@we.umg.edu.pl; Tel.: +48-694-458-333

**Keywords:** autonomous navigation, automatic radar plotting aid, safe objects control, game theory, computer simulation

## Abstract

This article provides an analysis of the autonomous navigation of marine objects, such as ships, offshore vessels and unmanned vehicles, and an analysis of the accuracy of safe control in game conditions for the cooperation of objects during maneuvering decisions. A method for determining safe object strategies based on a cooperative multi-person positional modeling game is presented. The method was used to formulate a measure of the sensitivity of safe control in the form of a relative change in the payment of the final game; to determine the final deviation of the safe trajectory from the set trajectory of the autonomous vehicle movement; and to calculate the accuracy of information in terms of evaluating the state of the control process. The sensitivity of safe control was considered in terms of both the degree of the inaccuracy of radar information and changes in the kinematics and dynamics of the object itself. As a result of the simulation studies of the positional game algorithm, which used an example of a real situation at sea of passing one's own object with nine other encountered objects, the sensitivity characteristics of safe trajectories under conditions of both good and restricted visibility at sea are presented.

## 1. Introduction

The subject of this article directly concerns sensors, which are the basic part of vessel detection and navigation in the process of ensuring safe control of marine objects. Sensors, such as radars, logs and gyro-compasses, form a source of information for the Automatic Radar Plotting Aid ARPA system, which are mandatory pieces of equipment for each ship to prevent collisions. However, its functional scope is limited to the determination of the safe maneuver of the object to the most dangerous object encountered, followed by its simulation in an accelerated time scale [1,2,3]. Modern navigation systems aim to use computer decision support systems that take many factors into account [4,5,6,7,8,9].

First, we take into account the subjectivity of the navigator in the assessment of the situation as well as the kinematics and dynamics of the objects encountered [10]. According to Lloyd Register statistics, the human errors caused by subjectivity account for about 60% of the causes of maritime accidents [11,12].

Secondly, we take into account the game nature of the anti-collision process, which results from the imperfection of maritime law and the complexity of the actual navigational situation at sea [13].

The influence of the information accuracy from sensors on the current state of the transport process and on the quality of safe control, which can be determined through the analysis of the control sensitivity to information inaccuracy, becomes important [14,15,16,17]. Most of the scientific literature concerns the sensitivity analysis of deterministic systems [18,19,20,21]. Therefore, the purpose of this study is to conduct the sensitivity analysis of the game control system for safe moving objects.

The process of managing the autonomous marine vehicles as a complex dynamic control object depends both on the accuracy of the measurements determining the current navigational situation, which was measured using the devices of the automatic radar plotting aids (ARPA) anti-collision system, and on the mathematical model of the process used to synthesize the object control algorithm. 

The ARPA system allows us to track the automatically encountered object *j*th by determining its motion parameters, including velocity *V_j_* and course ψ*_j_*, and approach elements to its own ship. Furthermore, we can also determine *D_jmin_* (distance of the nearest approach point, DCPA_j_) and *T_jmin_* (time to the nearest approach point, TCPA_j_) (Figure 1).

In theory and practice, there are many methods for determining a safe maneuver or the safe trajectory of the own object while passing other objects. The simplest method is to determine the course change maneuver or the speed of the own object in relation to the most dangerous object encountered. 

In one article [22], the “time to safe distance” upon the detection of dangerous objects was proposed as a potentially important parameter, which should be accompanied by a display of possible evasive maneuvers. The acceptable solutions for altering the course range should comply with the international regulations for preventing collisions at sea (COLREGs), as presented in [23].

The most important purpose of the control process is to determine a certain sequence of maneuvers in the form of a safe trajectory of the own object. The safe trajectory of the own object can be distinguished in deterministic terms without considering the maneuvering of other objects encountered. In the game approach, this is based on the use of a cooperative or non-cooperative game model of the control process [24,25].

The safety distance D_s_ of the passing objects, which was subjectively determined by the navigator in the current navigational situation, is important for safe navigation. This value depends on the current state of visibility at sea, which is classified by the international rules for preventing collisions of ships at sea (COLREGs) as either good or restricted visibility at sea.

Therefore, the aim of this study is to assess the sensitivity of the quality of security checks and games under the conditions of good and limited visibility at sea.

## 2. The Safe and Game Object Control in Autonomous Ship Navigation

The complexity of the situation when many dynamic objects are passed at sea provides the possibility of using a game model for the control process, given the fairly limited rules of international law of the sea route (COLREGs) in terms of good and limited visibility at sea and a large influence of the navigator's subjectivity in making the final maneuvering decision. It is possible to describe the process in the form of static positional and matrix games or in the form of dynamic differential games. This article proposes the use of the positional game, which is the most appropriate for this type of control process.

The basis of the positional game involves assigning the maneuver strategy of the own object to the current positions of *p*(*t_k_*) objects in the current step *k*. As such, the process model considers all possible changes in the course and speed of the encountered objects during the control [26,27,28,29,30,31,32,33,34,35,36,37,38,39,40,41,42] (Figure 2).

The process state is determined by the coordinates of the object's own position and the positions of the objects encountered as follows:(1)$$\begin{array}{l} x_{0}=\left(X_{0},\;Y_{0}\right)\;,\;x_{j}=\left(X_{j},\;Y_{j}\right)\\ j=1,\;2,\;...,\;J \end{array}\}.$$

The control algorithm generates the own object's movement strategy at the present time t_k_, based on information from the ARPA anti-collision system on the relative position of the objects that it meets:(2)$$p(t_{k})=\left[\begin{array}{l} x_{0}(t_{k})\\ x_{j}(t_{k}) \end{array}\right]\;\;\;j=1,\;2,...,\;J\;\;\;\;\;\;\;k=1,\;2,...,\;K.$$
Thus, in the multi-stage positional game model, at each discrete time *t_k_*, the own object knows the positions of the objects encountered.

The following navigation restrictions are imposed on the components of the process state, which consist of the acceptable coordinates of the own position and the objects encountered:(3)$$\left\{x_{0}\left(t\right),\;x_{j}\left(t\right)\right\}\;\;\in P.$$

The limits of the control values of the own and met objects are determined as: (4)$$u_{0}\in U_{0}\;,\;u_{j}\in U_{j}\;\;\;\;\;\;\;j=1,\;2,...,\;J.$$
which considers the ship movement kinematics, recommendations of the COLREGs rules and the conditions required to maintain a safe passing distance:(5)$$D_{j\min}=\min D_{j}(t)\geq D_{s}.$$

The sets of acceptable maneuvering strategies of the own object *U*_0*j*_(*p*) and the met objects *U_j_*_0_(*p*) are dependent, which means that the choice of control *u_j_* by the *j*th object changes the sets of acceptable strategies of other objects.

Figure 3 shows the geometrical structure of the sets of acceptable safe strategies for the own object and for one met *j*th object. First, the value of the safe ships passing at a distance *D_s_* is assumed. After this, they are positioned to be at a tangent to circles with a radius *D_s_*, which cut off the areas of safe possible changes in the courses and speeds of the own and the encountered objects in the districts with the *V* and *V_j_* rays.

The method for determining the total sets of acceptable safe strategies of the own object, while passing with many objects encountered simultaneously, is shown in Figure 4, which utilizes the example of passing the own object with six objects encountered at a safe distance *D_s_*.

The algorithm of the safe cooperative control of the own object *u*_0_*(*t_k_*) at each stage *k* is implemented using the following three tasks:

(1)Task min1: determine the safe admissible and optimal control of the own object *u*_0*j*_ in relation to the encountered *j*th object from the appropriate set of acceptable strategies *U*_0*j*_*^PS^* or *U*_0*j*_*^SS^*, which must comply with the 15th rule of COLREGs and ensure the lowest loss of the path for the safe passage of the encountered object (point 1 in Figure 3), (2)Task min2: determine the safe admissible and optimal control of the encountered *j*th object *u_j_*_0_ in relation to the own object from the appropriate set of acceptable strategies *U*_j*0*_*^PS^* or *U_j_*_0_*^SS^*, which must comply with the 15th rule of COLREGs and ensure the lowest loss of the path for safe pasting of the own object (point 2 in Figure 3),(3)Task min3: determine the safe and optimal control of the own object u_0_ in relation to all *J* objects (point 3 in Figure 4) of the following dependence:

(6)$$D^{*}\left(x_{0}\right)=\underset{u_{0}}{\min3}\;\underset{u_{j0}}{\min2}\;\underset{u_{0j}}{\min1}\;D\left[x_{0}\left(t_{k}\right)\right],\;j=1,\;2,\;...,\;J.$$
where *D* is the distance of the own object to the nearest point of return *P*_0*r*_ on the reference route.

The criterion for choosing the best trajectory of the own object is to calculate such course values and the speed, which would provide the smallest loss of the path for the safe passage of the encountered objects at a distance that is no less than the value of *D_s_* previously accepted by the navigator.

Through the three-fold use of the *linprog* function, which is linear programming from the MATLAB Optimization Toolbox software, a cooperative multi-stage Positional Game (PG) algorithm was developed to determine the safe trajectory of the own object.

## 3. Control Sensitivity Analysis

The sensitivity analysis refers to the identification of the static and dynamic properties of control objects and to the synthesis of automatic control systems, particularly optimal, adaptive and game systems. A distinction is made between the sensitivity of the object model itself or the control process of changes in its operating parameters and the sensitivity of the optimal, adaptive or game control. This is both in terms of changes in parameters and the influence of disturbances, and impacts of other objects. Therefore, the *s_x_* sensitivity functions of the optimal control u of the game process described by the state variables *x* can be represented as the following partial derivatives of quality control index *Q*:(7)$$s_{x}=\frac{\partial Q\left[x\left(u\right)\right]}{\partial x}.$$

The game control quality index *Q* acts as the form of payment for the game, which consists of integral payments and the final payment:(8)$$Q=\int_{t_{0}}^{t_{K}}{\left[x\left(t\right)\right]}^{2}+r_{j}\left(t_{K}\right)+d\left(t_{K}\right).$$

The integral game payment represents the loss of the path through its own object when passing the objects encountered and the final payment determines the final collision risk *r_j_*(*t_k_*) with respect to the *j*th object encountered and the final deviation of the trajectory of the object *d*(*t_k_*) from the reference trajectory. 

Testing the sensitivity of game control will complete the sensitivity analysis of the final game payment *d*(*t_k_*):(9)$$s_{i}=\frac{\partial d\left(t_{K}\right)}{\partial x_{i}}.$$

Considering the practical application of the game control algorithm for the own object in a collision situation, it is recommended that the sensitivity analysis of a secure control should be conducted in terms of the information accuracy obtained from the ARPA anti-collision radar system in the current situation and in relation to changes in the kinematic parameters and dynamic control.

The permissible average errors that may be caused by an anti-collision system sensors may have the following values for:radar: bearing ±0.25° and distance ±0.05 nm,gyrocompass: ±0.5°,log: ±0.5 kn.

The algebraic sum of all errors affecting the image of the navigational situation cannot exceed ±5% for absolute values and ±3° for angular quantities.

### 3.1. Sensitivity of Safe Ship Control to Inaccuracy of Information from Sensors of ARPA System

*SP* represents such a set of information about the control of the State Process in a navigational situation:(10)$$SP=\{V,\psi ,V_{j},\psi_{j},D_{j},N_{j}\}.$$

*SP_e_* represents a set of information from the sensors of ARPA system, which contains errors in measurement and processing parameters:(11)$$SP_{e}=\{V\pm \delta V,\psi \pm \delta \psi ,V_{j}\pm \delta V_{j},\psi_{j}\pm \delta \psi_{j},\;D_{j}\pm \delta D_{j},N_{j}\pm \delta N_{j}\}.$$

The relative sensitivity of the final payment in the *s_x_* game as the final deviation of the safe trajectory of the ship *d_k_* from the reference trajectory is expressed as follows:(12)$$s_{x}=\left|\frac{d_{K}(SP_{e})-d_{K}(SP)}{d_{K}(SP)}\right|100\%.$$

(13)$$s_{x}=\{s_{V},s_{\psi },s_{Vj},s_{\psi j},s_{Dj},s_{Nj}\}.$$

### 3.2. Sensitivity of Safe Own Object Control to Autonomous Navigation Process Parameter Alterations

*PP* is a set of state Parameter Processes of control, which is expressed as follows:(14)$$PP=\{t_{m},D_{s},\Delta t_{k},\Delta V\}.$$

*PP_e_* represents a set of parameters containing errors in measurement and processing parameters:(15)$$PP_{e}=\{t_{m}\pm \delta t_{m},D_{s}\pm \delta D_{s},t_{k}\pm \delta t_{k},\Delta V\pm \delta \Delta V\}.$$

The relative sensitivity of the final payment in the game *s_p_*, which represents the final deviation of the safe trajectory of the ship *d_K_* from the assumed trajectory, will be:(16)$$s_{p}=\left|\frac{d_{K}(PP_{e})-d_{K}(PP)}{d_{K}(PP)}\right|100\%.$$
(17)$$s_{p}=\{s_{tm},s_{Ds},s_{Ts},s_{\Delta tk},s_{\Delta V}\}.$$
where *t_m_* is the advance time of the maneuver with respect to the dynamic properties of the own ship, *t_k_* is the duration of one stage of the ship's trajectory, *D_s_* is the safe distance and *T_s_* is the safe time of the approach.

## 4. Sensitivity Characteristics

The computer simulation of the PG algorithm, which represents the computer software supporting the navigator's maneuvering decision, was conducted using an example of a real navigational situation of the *J* = 9 objects encountered.

### 4.1. Sensitivity Characteristics of Game Own Object Control in Good Visibility at Sea

The safe trajectory of the own object and sensitivity characteristics, which were determined by the PG algorithm in the MATLAB/Simulink software, are presented in Figure 5 and Figure 6.

### 4.2. Sensitivity Characteristics of Game Own Object Control in Restricted Visibility at Sea 

The safe trajectory of the own object and sensitivity characteristics, which were determined by the PG algorithm in the MATLAB/Simulink software, are presented in Figure 7 and Figure 8.

## 5. Conclusions

The use of simplified models of a dynamic process game for the synthesis of optimal control allowed us to determine the safe trajectories of an own object in situations that involve passing a large number of encountered objects in a certain course sequence and speed maneuvers. 

The developed algorithms also consider the COLREGs rules and maneuver advance time in addition to estimating the object's dynamic properties and assessing the final deviation of the actual trajectory from the reference value.

The following conclusions follow from the course of the sensitivity characteristics presented in Figure 5, Figure 6, Figure 7 and Figure 8:the sensitivity characteristics are nonlinear, with most being angular values,in the range of the most important changes in input quantities (1% for absolute values, 1 degree for angular quantities and 20% for process parameters), the sensitivity usually does not exceed 20%,together with the deterioration of visibility at sea, the sensitivity to the inaccuracy of information about the object encountered and the change in the value of safe distance and time of approach increases.

The considered control algorithms are the formal models of the navigator's thinking process that controls the objects’ movement and maneuvering decisions. Therefore, they can be used in the construction of a new model of the ARPA system containing a computer that supports the decision-making of the navigator.

## Figures and Tables

**Figure 1 sensors-19-01816-f001:**
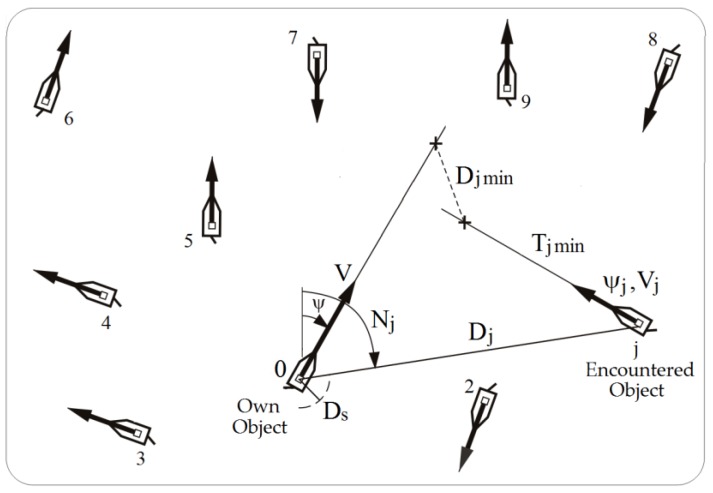
The navigational situation of the passage of the own object 0 moving at speed *V* and course ψ with the *j*th object encountered when moving at speed *V_j_* and course ψ*_j_*. In this figure, *D_j_* is distance, *N_j_* is bearing and *D_s_* is safe distance.

**Figure 2 sensors-19-01816-f002:**
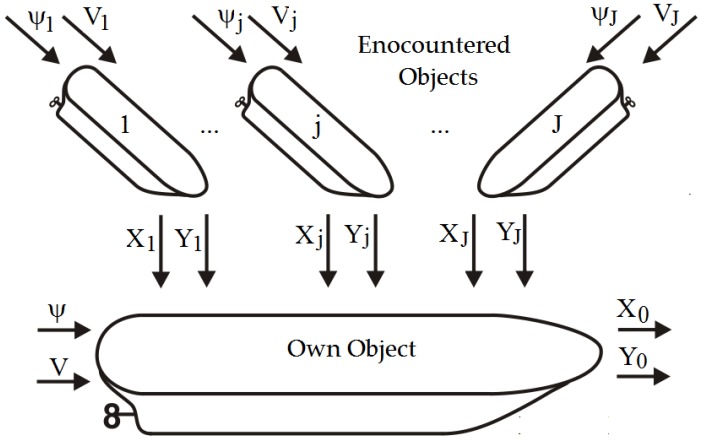
Block diagram of the positional game model in the situation with passing objects.

**Figure 3 sensors-19-01816-f003:**
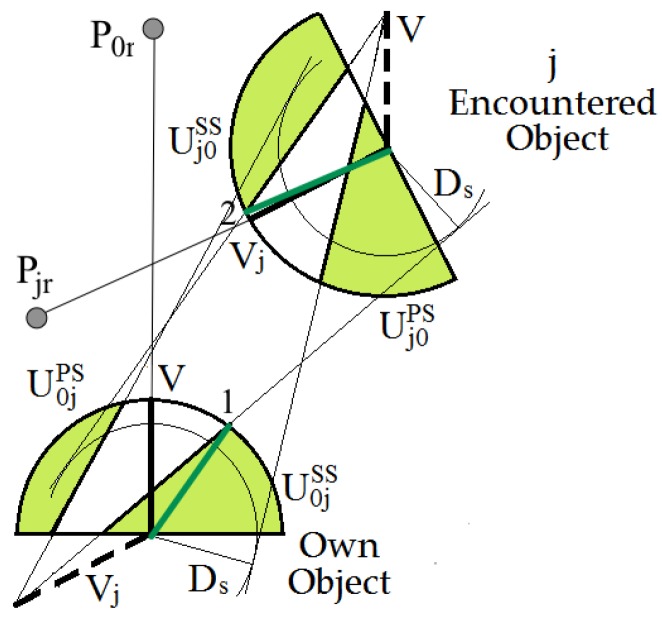
The method of determining the acceptable strategy sets of the own object *U*_0*j*_ = *U*_0*j*_*^PS^* ∪ *U*_0*j*_*^SS^* and the *j*th encountered object *U_j_*_0_ = *U_j_*_0_*^PS^* ∪ *U_j_*_0_*^SS^* for the port side (PS) and starboard side (SS). In this figure, *P*_0*r*_ is the turning point of the rhumb line of the own object, *P_jr_* is the turning point of the rhumb line of the encountered object, 1 is a safe maneuver for changing the course of the own object and 2 is a safe maneuver for changing the course of the encountered object.

**Figure 4 sensors-19-01816-f004:**
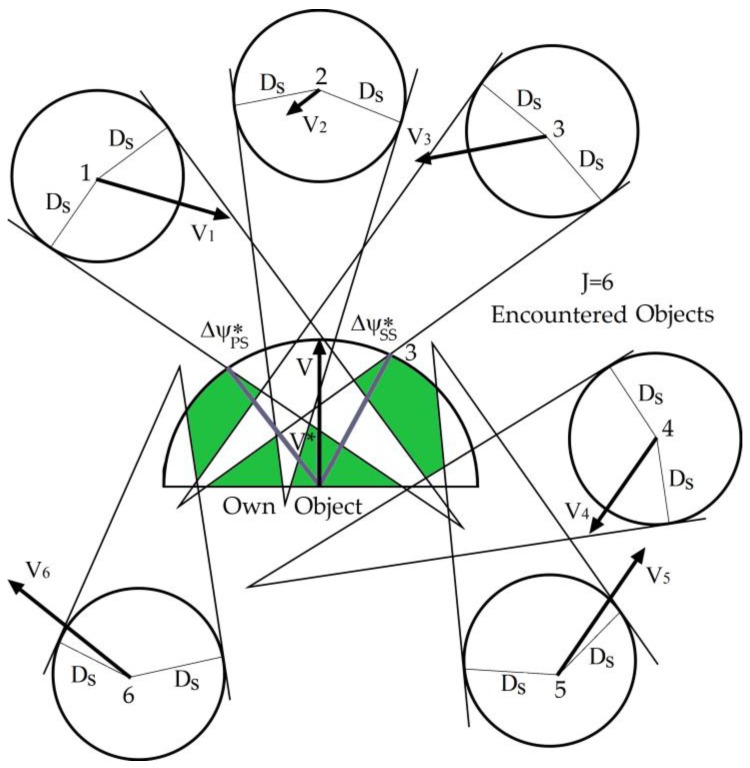
Areas of acceptable safe strategies of the own object in relation to the six objects encountered: 3 is the optimal maneuver for changing the course of the own object *u*_0_* = Δψ**_SS_* when safely passing the six met objects.

**Figure 5 sensors-19-01816-f005:**
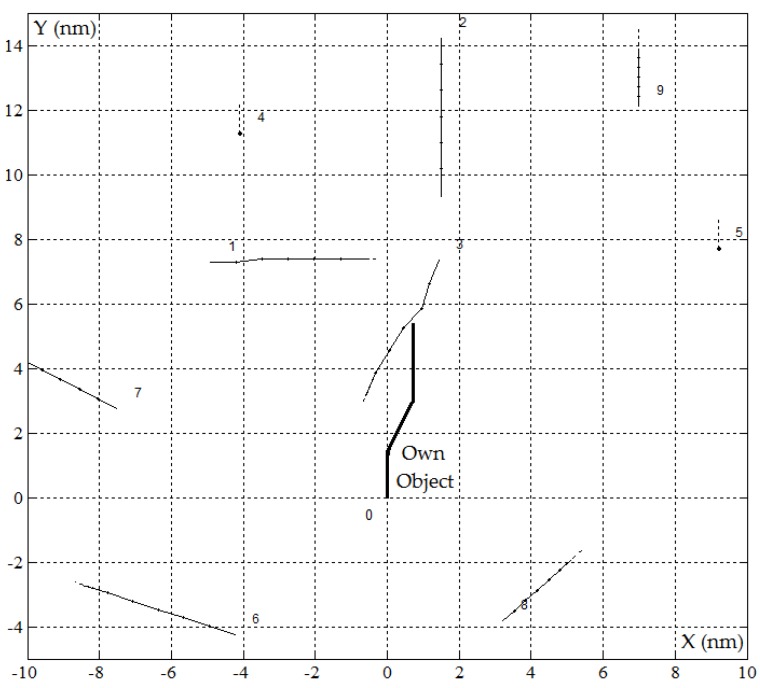
The safe trajectory of the own object for the positional game PG_gv algorithm in good visibility at sea where *D_s_* = 0.5 nm in the situation of passing *J* = 9 encountered objects, *r*(*tK*) = 0 and *d*(*tK*) = 0.71 nm.

**Figure 6 sensors-19-01816-f006:**
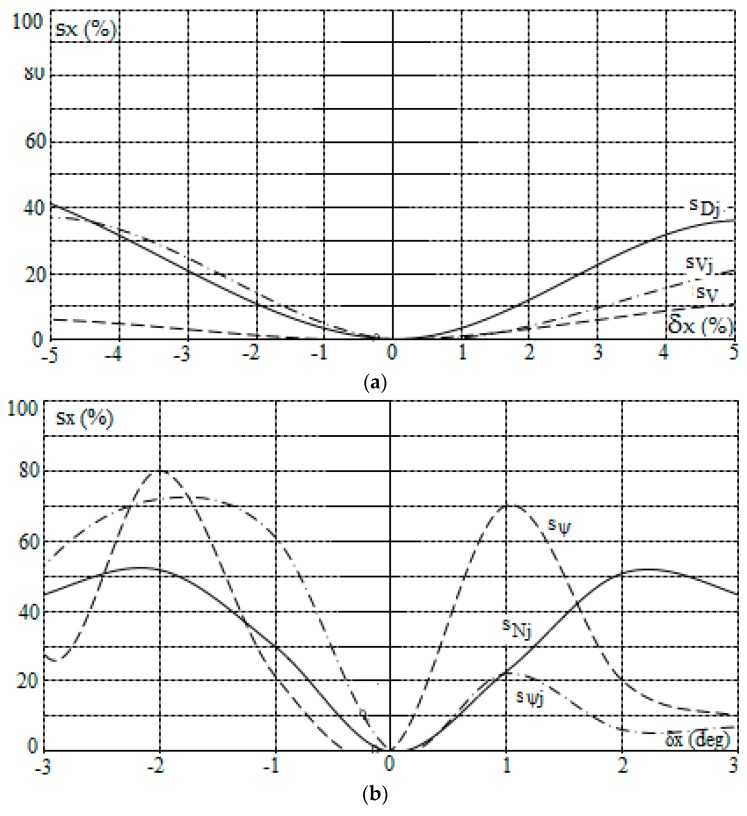
Sensitivity characteristics of the positional game control of the own object in good visibility at sea according to PG_gv algorithm as a function of: (**a**) absolute values of the information from sensors, (**b**) angular values of the information from sensors and (**c**) values of the control process parameters.

**Figure 7 sensors-19-01816-f007:**
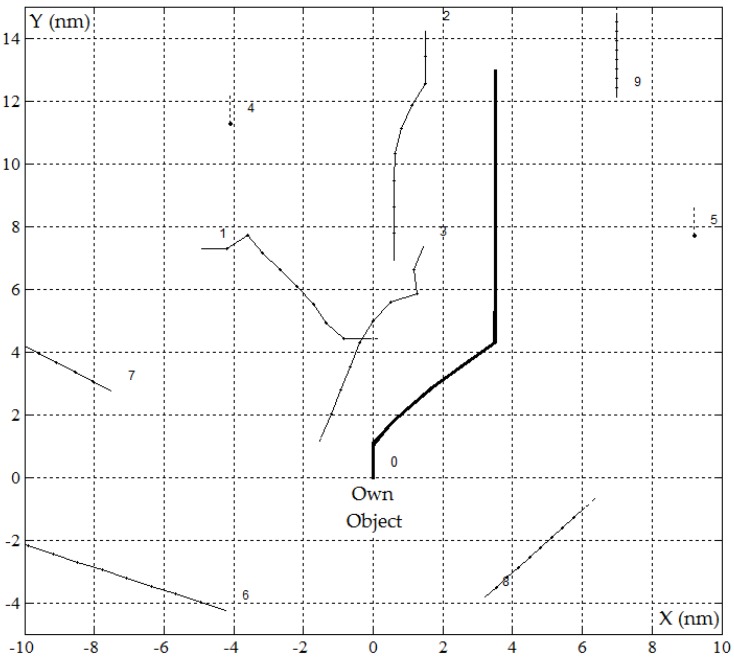
The safe trajectory of the own object for positional game PG_rv algorithm in restricted visibility at sea where *D_s_* = 1.5 nm in the situation of passing *J* = 9 encountered objects, *r*(*tK*) = 0 and *d*(*tK*) = 3.47 nm.

**Figure 8 sensors-19-01816-f008:**
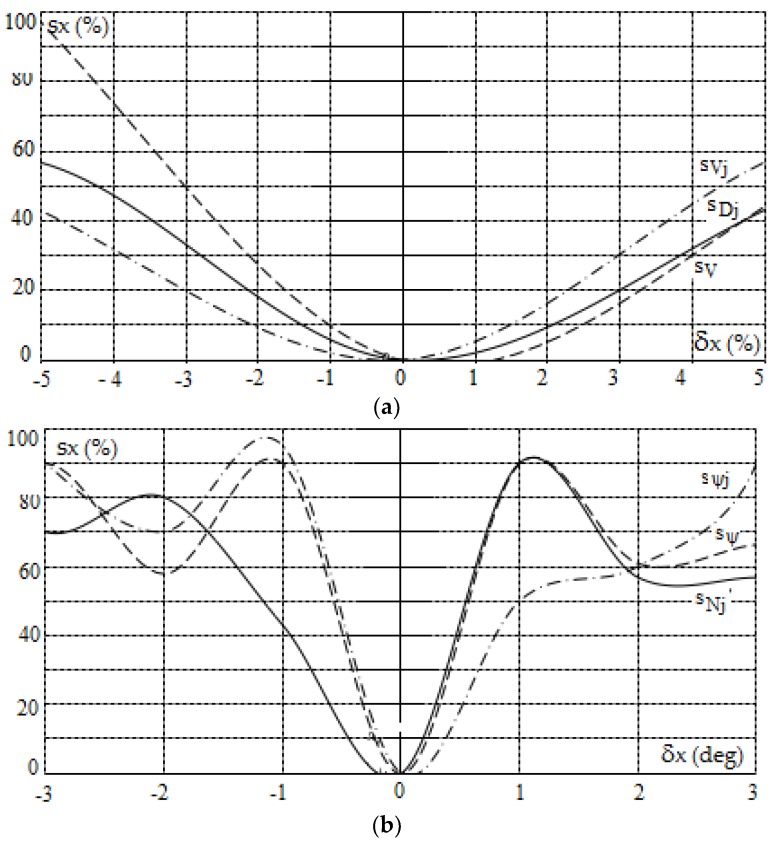
Sensitivity characteristics of the positional game control of the own object in restricted visibility at sea according to the PG_rv algorithm as a function of: (**a**) absolute values of the information from sensors, (**b**) angular values of the information from sensors and (**c**) values of the control process parameters.

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
