# Peer review of "Sensitivity of Safe Trajectory in a Game Environment on Inaccuracy of Radar Data in Autonomous Navigation"

_sensors, 2019, doi:10.3390/s19081816_

Round 1
Reviewer 1 Report
I am confident that the article is interesting and essential in terms of research. The mathematical background is very well substantiated and supported. The author shows off significant experience in the topic. In order to enhance the article quality, I suggest the following remarks be taken into account:
1. Introduction: It is good idea to add a few sentences about another solutions to the problem of preventing ship collisions at sea (e.g Lenart equations). The author should enrich their text with relevant references. For example, as:
- „Analysis of Collision Threat Parameters and Criteria" Journal of Navigation vol. 68, no. 5, 2015 (887-896)
- „Presentation algorithm of possible collision solutions in a navigational decision support system” Scientific Journals of the Maritime University of Szczecin no. 38(110), 2014 (20-26)
2. Figure 1: Please provide the meaning of notation Ds.
3. Line 68: necessity - seems to be a little too strong a statement.
4. Equation 2: Vector not marked in bold and consequently further.
5. Line 96: Figures 3-4 seems to be insufficiently described.
6. Line 110: What regulations have been taken into account (COLREGs)?
7. Conclusions: Please refer to figures 5-8.
Author Response
I have included all the comments of the Reviewer 1 in the article text.

Reviewer 2 Report
The theme is not new. Subject is not entirely relevant for SENSORS. Article approaches to other editions of publishing house. Models justified truncated. It is necessary to improve the References review. It is necessary to focus more on the subject of the journal.
The review of Inrtoduction should be expanded with additional
information. A very brief overview on modern systems for navigation. It
is imperative for the industry to consider security issues focusing on
the requirements of the registry and IMO.
There is no modern analysis of literary sources in sufficient volume.
Show how these studies affect safety.
The article should also be more closely linked to the modern journal.
It is necessary to deliver information from the Loyd Register on the accident rate of ships.
It is necessary to reflect the prospects for the implementation and development of these studies.
Author Response
I took into account all the comments of the Reviewer 2 in Introduction, by inserting a supplementary text and in References by inserting new dozen literature items directly related to sensors and sensitivities.

Reviewer 3 Report
In this article an analysis of the accuracy of safe control in game conditions of cooperation of objects during maneuvering decisions was made. A method for determining safe object strategies based on a cooperative multi-person positional modeling game is presented. The validility of the method was verified by the simulation.
Generally, this article has certain publishing value, but the obscure English expression makes the article difficult to understand, every sentence is too long. So I suggest all English expressions should be revised by native speakers.
Author Response
Article has undergone English language editing by MDPI. The text has been checked for correct use of grammar and common
technical terms, and edited to a level suitable for reporting research in a scholarly journal.
MDPI uses experienced, native English speaking editors.

Round 2
Reviewer 2 Report
All done well
Reviewer 3 Report
This article provides an analysis of the autonomous navigation of marine objects and an analysis of the accuracy of safe control in game conditions of the cooperation of objects during maneuvering decisions. A method for determining safe object strategies based on a cooperative multi-person positional modeling game is presented. The validity of the method was verified by the simulation.
After English language editing, the article has reached the level to been published in sensors journal.